# Outcomes of open versus single-incision laparoscopic totally extraperitoneal inguinal hernia repair using propensity score matching: A single institution experience

**Kanghee Lee, Jin Ho Lee, Soomin Nam, Jae Uk Chong, Hyung Soon Lee** *

Department of Surgery, National Health Insurance Service Ilsan Hospital, Goyang, Korea

* soon0925@nhimc.or.kr

**Data Availability Statement:** All relevant data are within the manuscript and its Supporting information files.

## Abstract

### Purpose

The aim of the current study was to compare the outcomes between open and single-incision laparoscopic totally extraperitoneal (SILTEP) inguinal hernia repair.

### Methods

To compare the outcomes between the open and SILTEP groups, we performed propensity score matching to adjust for significant differences in patient characteristics. The outcomes were compared between the matched groups.

### Results

Record review identified 477 patients who had undergone inguinal hernia repair from November 2016 to November 2018. Seventy-one patients were excluded from the propensity score matching because of age <18, femoral hernia, conventional 3-port laparoscopic repair, incarcerated hernia, and combined operation. SILTEP in 142 and open repair in 264 patients were identified. After propensity score matching, these individuals were grouped into 82 pairs. Spinal anesthesia was administered more often in the open group than in the SILTEP group. Operation time was significantly longer in the SILTEP group than in the open group (49.6 ± 17.4 vs. 64.8 ± 28.4 min, p < 0.001). However, urinary retention rates of the open group were significantly higher than that of the SILTEP group (11.0% vs. 0%, p = 0.003). The SILTEP group showed significantly lower pain scores at postoperative 6, 12, and 24 hours, and significantly lower rates of intravenous analgesic requirements through postoperative day 1 (30.5% vs. 13.4%, p = 0.008) compared with the open group.

### Conclusion

The outcomes of SILTEP repair were comparable to those of open repair. SILTEP repair may have advantages over open repair for reducing immediate postoperative pain (≤24 hours).

**Funding:** This study was supported by a faculty research grant from the National Health Insurance Service Ilsan Hospital (NHIMC 2019CR027). The funders had no role in study design, data collection and analysis, decision to publish, or preparation of the manuscript.

**Competing interests:** NO authors have competing interests.

# Introduction

During the last two decades, inguinal hernia surgery sought to reduce surgical trauma by reducing access and to improve clinical outcomes [1]. Therefore, laparoscopic inguinal hernia repair has gained wide acceptance and offers promising outcomes [2,3]. Although laparoscopic repair has shown excellent results, surgeons have sought to enhance its benefits by applying single-incision laparoscopic surgery in inguinal hernia repair. Consequently, single incision laparoscopic procedures are a rapidly evolving trend and are used increasingly in inguinal hernia repair [4,5].

Annually, an average of 34,604 inguinal hernia repairs are performed in Korea. Laparoscopic inguinal hernia repairs dramatically increased from 2.4% in 2007 to 29.5% in 2015, whereas the total number of inguinal hernia surgery cases remained relatively constant [6]. Similar to the global trend of using minimal invasive surgery for hernia, single-incision laparoscopic inguinal hernia repair has also recently gained popularity in Korea [7–10]. However, more than 70% of Korean surgeons still perform open repair. And conventional laparoscopic totally extraperitoneal (CTEP) repair is preferred to single-incision laparoscopic repair because the latter is associated with a steeper learning curve and a higher intra-operative level of difficulty due to instrument collision and loss of the required triangulation [5].

The first case of single-incision laparoscopic totally extraperitoneal (SILTEP) inguinal hernia repair was reported in 2008 [11]. The safety and efficacy of SILTEP repair has been assessed in various prospective studies [4,5]. In most of the studies comparing CTEP and SILTEP repair, there were no significant differences observed between the two groups with regard to postoperative hospital stay, complications, recurrences, and postoperative pain scores. However, there have been no reports comparing the outcomes of open versus SILTEP inguinal hernia repair. Thus, the aim of the current study was to compare the outcomes between open and SILTEP inguinal hernia repair.

# Material and methods

## Patient selection

Medical records were retrospectively reviewed to identify patients who underwent surgery for inguinal hernia at National Health Insurance Service Ilsan Hospital between November 2016 and November 2018. Exclusion criteria were as follows: (1) under 18 years of age, (2) underwent CTEP repair, (3) emergency operation for treatment of acute bowel incarceration, (4) femoral hernia, and (5) combined operation with inguinal hernia repair. The current study was performed after approval by the Institutional Review Boards of National Health Insurance Service Ilsan Hospital (NHIMC2019-03-004). This was a retrospective observational study; therefore, the requirement for informed consent was waived by the Institutional Review Boards. The raw and analysis-ready datasets were anonymized by removing all personally identifiable information.

## Review of patient data

The patient characteristics included age, sex, height, weight, body mass index (BMI), side of hernia, type of hernia, recurrent hernia, American Society of Anesthesiologists (ASA) score, duration of symptoms, type of anesthesia, contents of hernia sac, and previous lower abdominal surgery history. Surgical outcomes included operative time, length of hospital stay, postoperative pain score, postoperative complications, and postoperative intravenous analgesic usage. Postoperative complications were categorized into seroma, hematoma, urinary retention, wound infection, bladder injury, recurrence, and chronic pain. Intraoperatively, conversions

from SILTEP to open repair were documented. The operative time was recorded as the time from incision to application of dressing. Urinary retention was defined as a case requiring nela-ton or foley insertion for voiding. Postoperative pain score was measured using the visual ana-log scale (VAS) at 6, 12, and 24 h postoperation. Chronic pain was defined as pain persisting in the groin for more than three months following hernia repair [1]. A recurrence was defined as unambiguous bulging of the abdominal wall along the course of the inguinal canal with reposi-tion at relaxation confirmed using ultrasonography or computed tomography.

## Surgical technique

In SILTEP repair, under general anesthesia, a single 25-mm umbilical incision was made fol-lowed by dissection of the subcutaneous tissue down to the rectus abdominis sheath. The ante-rior sheath was opened with an incision approximately 3 cm in length, and blunt dissection using a finger or gauze was performed between the rectus muscle and the posterior sheath. A disposable Lapsingle® (Sejong Medical, Ltd, Gyeonggi-do, South Korea) trocar was inserted in front of the posterior rectus sheath. After 8 mmHg pressure of $CO_2$ insufflation through the insufflation channel of the port, an endoscopic camera 5 mm in diameter and 45 cm in length was inserted. Graspers and monopolar diathermy were used to dissect the preperitoneal space through two 5-mm channels. The surgical procedure was performed after the patient was placed in the Trendelenburg position, with the side of hernia tilted up. Lateral space was dis-sected toward the anterior superior iliac spine and continued medially until the inferior epigas-tric vessels, symphysis pubis, and cord structures were identified. In cases of direct hernia, the pseudo-sac of the transversalis fascia was anchored on the pubic bone by tacks (Tacker, Covi-dien plc, Dublin, Ireland). In cases of indirect hernia, the hernia sac was isolated, freed from the spermatic cord, and reduced from the internal ring by gentle traction and dissection. Indi-rect hernia sacs were routinely ligated just beyond the internal ring with a Vicryl Endoloop® (Ethicon, Somerville, NJ) and divided using endo-scissors. A 3DMax® (Bard, Murray Hill, NJ, USA) mesh was placed around the spermatic cord from the pubic symphysis to the anterior iliac spine laterally without fixation. The space was deflated under direct visualization without drainage. The incision wound was repaired with absorbable 3–0 sutures using subcuticular methods (Fig 1) [7]. Open repairs were performed under spinal or general anesthesia accord-ing to the standard Lichtenstein tension-free technique [12].

## Patient follow-up

The patients were discharged after surgery without specific restrictions with regard to mobili-zation or exercise. Patients were seen in the outpatient clinic for regular follow-up at 1 week postoperatively. Additional follow-up visits to the outpatient clinic were determined according to the patient's postoperative clinical course.

## Statistical analysis

To compare the outcomes between the open and SILTEP groups, we performed propensity score matching to adjust for significant differences in patient characteristics. The propensity scores were estimated by multiple logistic regression analysis. Regression analysis predicted the probability that each patient would be treated based on six covariables: age, sex, BMI, ASA score, bilateral, and recurrent hernia. Using these six covariables in the regression analysis, a propensity score was calculated for each patient. The discrimination and calibration abilities of the propensity score models were assessed using the C-statistic and the Hosmer-Lemeshow statistic [13]. The model was then used to obtain a one-to-one match for the open group and SILTEP group. Finally, each patient with SILTEP repair was matched to one patient with open

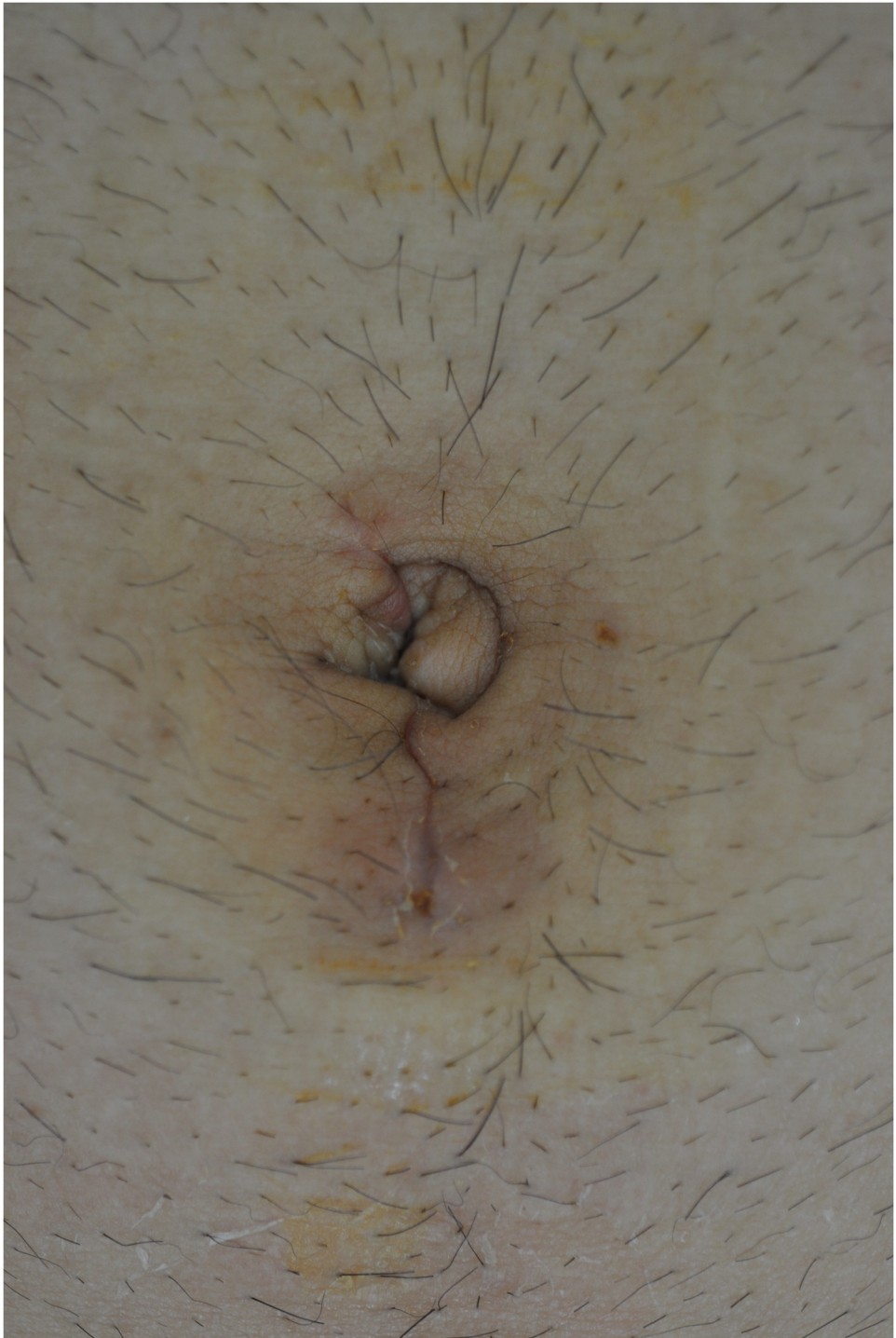

**Fig 1. Postoperative scar at 1 week after SILTEP repair.**

repair with an identical propensity score. After the propensity score-matched sample was formed, we assessed baseline variable balance between the two propensity-matched cohort groups. Continuous variables were compared using the paired t-test or the Wilcoxon signed-rank test, as appropriate, and categorical variables were compared using the McNemar's or

marginal homogeneity test, as appropriate [14]. Propensity score matching was performed using SAS (version 9.2; SAS Institute Inc., Cary, NC). Data are presented as the number of patients (percentage) or as the mean ± standard deviation. We analyzed categorical variables using the Chi-square test or Fisher's exact test and continuous variables using the Mann-Whitney U test. P-values <0.05 were considered statistically significant. Statistical procedures were conducted using the statistical software SPSS 18.0 (SPSS Inc., Chicago, IL, USA).

## Results

### Patient characteristics

Record review identified 477 patients who had undergone inguinal hernia repair between November 2016 and November 2018. Seventy-one patients were excluded: age < 18 (n = 35); femoral hernia (n = 6); CTEP repair (n = 19); incarcerated hernia (n = 6); and combined operation (n = 5) (Fig 2). From the remaining 406 patients, SILTEP and open repairs were identified in 142 and 264 patients, respectively. After propensity score matching, these individuals were balanced into 82 pairs (Table 1). The two groups showed significant differences in age, ASA score, and recurrent hernia before matching. These differences between the groups were eliminated after matching. There were no significant differences between the two groups with regard to history of previous lower abdominal surgery, duration of symptoms, site of hernia, type of hernia, and hernia sac content (Table 2). However, the open group showed a significantly higher proportion of patients that received spinal anesthesia than those that received general anesthesia.

### Comparison of the outcomes between the open and SILTEP repair groups

Operation time was significantly longer in the SILTEP group than in the open group (49.6 ± 17.4 vs. 64.8 ± 28.4 min, p < 0.001) (Table 3). The SILTEP group showed significantly

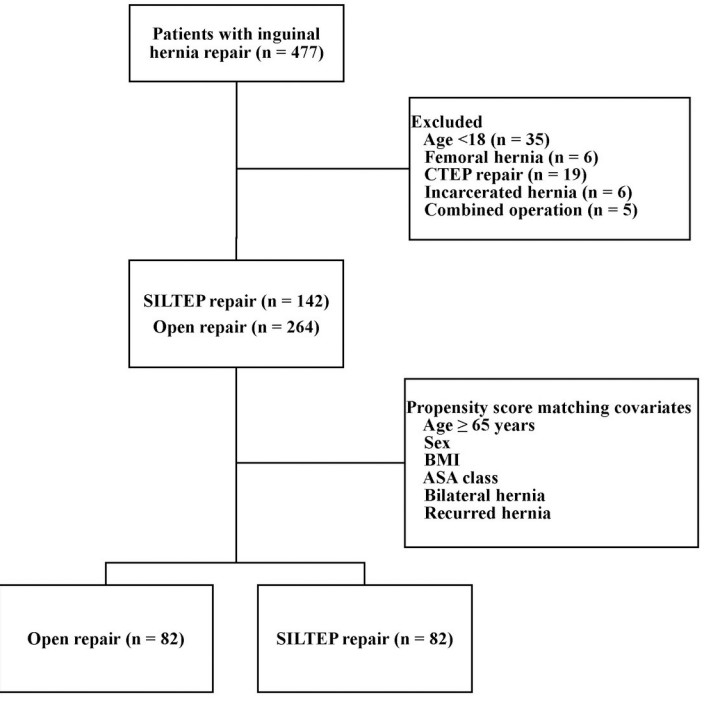

**Fig 2. Flowchart of patient selection.** CTEP, conventional laparoscopic totally extraperitoneal; SILTEP, single-incision laparoscopic totally extraperitoneal; BMI, body mass index; ASA, American Society of Anesthesiologists.

**Table 1. Before and after propensity score matching of baseline variables in the SILTEP group versus the open group.**

| Variable | Category | Before Matching | | | After Matching | | |
|---|---|---|---|---|---|---|---|
| | | SILTEP (n = 141) | Open (n = 264) | *P*-value | SILTEP (n = 82) | Open (n = 82) | *P*-value |
| Age ≥ 65 years | | 57 (40.4%) | 158 (59.8%) | < 0.001 | 42 (51.2%) | 46 (56.1%) | 0.531 |
| Female sex | | 10 (7.1%) | 14 (5.3%) | 0.468 | 2 (2.4%) | 2 (2.4%) | 1.000 |
| BMI [a] | | 23.56 ± 2.88 | 24.07 ± 3.13 | 0.108 | 23.39 ± 2.68 | 23.49 ± 2.65 | 0.817 |
| ASA score | 1 | 54 (38.3%) | 57 (21.6%) | 0.001 | 21 (25.6%) | 19 (23.2%) | 0.987 |
| | 2 | 57 (40.4%) | 107 (40.5%) | | 36 (43.9%) | 37 (45.1%) | |
| | 3 | 29 (20.6%) | 96 (36.4%) | | 24 (29.3%) | 25 (30.5%) | |
| | 4 | 1 (0.7%) | 4 (1.5%) | | 1 (0.6%) | 1 (0.6%) | |
| Bilateral hernia | | 4 (2.8%) | 16 (6.1%) | 0.154 | 4 (4.9%) | 3 (3.7%) | 1.000 |
| Recurrent hernia | | 12 (8.5%) | 8 (3.0%) | 0.015 | 3 (3.7%) | 4 (4.9%) | 1.000 |

Values are indicated as number (percentage) unless indicated otherwise;

[a] values are mean ± standard deviation.

SILTEP, single incision laparoscopic totally extraperitoneal; BMI, body mass index; ASA, American Society of Anesthesiologists.

**Table 2. Patient characteristics.**

| | Open (n = 82) | SILTEP (n = 82) | *P*-value |
|---|---|---|---|
| Previous lower abdominal surgery | 2 (2.4%) | 0 (0%) | 0.497 |
| Duration of symptoms (months) [a] | 10.7 ± 41.1 | 26.4 ± 101.4 | 0.197 |
| Hernia side | | | 0.860 |
| Right | 47 (57.3%) | 44 (53.7%) | |
| Left | 32 (39.0%) | 34 (41.5%) | |
| Bilateral | 3 (3.7%) | 4 (4.9%) | |
| Hernia type | | | 0.129 |
| Indirect | 69 (84.1%) | 62 (75.6%) | |
| Direct | 12 (14.6%) | 14 (17.1%) | |
| Pantaloon | 1 (1.2%) | 6 (7.3%) | |
| Hernia sac contents | | | 0.296 |
| Omentum | 20 (24.4%) | 10 (12.2%) | |
| Small bowel | 4 (4.9%) | 0 (0%) | |
| Anesthesia | | | <0.001 |
| General | 26 (31.7%) | 82 (100%) | |
| Spinal | 56 (68.3%) | 0 (0%) | |

Values are indicated as number (percentage) unless indicated otherwise;

[a] values are indicated as mean ± standard deviation.

lower pain scores at postoperative 6, 12, and 24 hours (p < 0.001). In addition, the SILTEP group showed significantly lower rates of intravenous analgesic requirements through postoperative day 1 (30.5% vs. 13.4%, p = 0.008). The rates of postoperative complications between the two groups showed no statistically significant difference. However, urinary retention rates of the open group were significantly higher than those of the SILTEP group (11.0% vs. 0%, p = 0.003). Urinary retention was managed and resolved using nelaton or foley catheter insertion. The most common complication in the SILTEP group was postoperative seroma formation, but all cases were managed conservatively and resolved by the 1-month follow-up. Bladder injury was identified in one patient in the SILTEP group intraoperatively. Bladder

**Table 3. Surgical outcomes of groups.**

| | Open (n = 82) | SILTEP (n = 82) | *P*-value |
|---|---|---|---|
| Operative time (min) [a] | 49.6 ± 17.4 | 64.8 ± 28.4 | <0.001 |
| Length of hospital stay (days) [a] | 2.7 ± 1.7 | 2.7 ± 0.8 | 0.908 |
| Postoperative complications | 14 (17.1%) | 11 (13.4%) | 0.665 |
| Seroma | 4 (4.9%) | 8 (9.8%) | 0.369 |
| Hematoma | 2 (2.4%) | 1 (1.2%) | 0.560 |
| Urinary retention | 9 (11.0%) | 0 (0%) | 0.003 |
| Wound infection | 0 (0%) | 0 (0%) | 1.000 |
| Bladder injury | 0 (0%) | 1 (1.2%) | 0.316 |
| Postoperative pain score (VAS) [a] | | | |
| 6 hours | 3.7 ± 1.1 | 3.1 ± 0.3 | <0.001 |
| 12 hours | 3.0 ± 0.6 | 2.6 ± 0.5 | <0.001 |
| 24 hours | 2.7 ± 0.5 | 2.1 ± 0.4 | <0.001 |
| IV analgesic requirements | | | |
| postoperative day 0 | 25 (30.5%) | 11 (13.4%) | 0.008 |
| postoperative day 1 | 12 (14.6%) | 1 (1.2%) | 0.001 |
| IV analgesic use (number) [a] | 1.5 ± 0.6 | 1.1 ± 0.3 | 0.029 |
| Recurrence | 4 (4.9%) | 4 (4.9%) | 1.000 |
| Chronic pain | 4 (4.9%) | 1 (1.2%) | 0.367 |

Values are indicated as number (percentage) unless indicated otherwise;

[a] values are mean ± standard deviation.

SILTEP, single-incision laparoscopic totally extraperitoneal; VAS, visual analog scale; IV, intravenous.

repair was immediately performed and foley insertion was used for the first five postoperative days. The patient recovered without further complications. Conversion to open repair or use of additional trocars was not required in the SILTEP group. The rates of recurrence and development of chronic pain were not significantly different between the two groups.

## Discussion

The present study demonstrated that the outcomes of SILTEP repair were not significantly different from those of open repair. Although operation time was significantly longer in the SILTEP group, the SILTEP group showed significantly lower postoperative pain scores and intravenous analgesic requirements through postoperative day 1. These results suggest that SILTEP repair may offer comparable outcomes and reduce immediate postoperative pain compared with open repair.

SILTEP inguinal hernia repair is associated with certain demanding intra-operative technical challenges and a steep learning curve [9]. However, many previous studies have demonstrated the technical feasibility and clinical safety of CTEP repair compared to open repair [15]. Recent studies have also shown the technical feasibility and clinical safety of SILTEP repair compared to CTEP repair [5,10]. Consistent with previous studies, the current study showed no significant difference in the rates of postoperative complications between the two groups. Unfortunately, one patient in the SILTEP group developed intra-operative bladder injury. However, the patient with the bladder injury was the third case of SILTEP repair. This major complication can be attributed to the learning phase of SILTEP repair. In this regard, major complications rarely occurred after overcoming the learning curve and SILTEP repair was safely performed comparable to open repair.

The development of postoperative urinary retention following inguinal hernia repair may be affected by various predisposing factors [16]. However, the most important predisposing factor for postoperative urinary retention after an inguinal hernia repair is the method of anesthesia [1]. Laparoscopic procedures have been traditionally performed under general anesthesia due to the respiratory changes caused by pneumoperitoneum. Thus, the use of general anesthesia may be considered a disadvantage of laparoscopic inguinal hernia repair. However, spinal anesthesia also has the disadvantage of being associated with delayed ambulation caused by slow recovery of sensory and motor function, long recovery room time, as well as urinary retention. Those potential limitations could impact discharge after inguinal hernia repair [17]. Bakota et al. demonstrated that urinary retention in open inguinal hernia repair in adults is statistically less frequent in the general anesthesia group compared with that in the spinal anesthesia group [18]. Furthermore, an analysis from the Danish Hernia Database found a higher incidence of medical complications in patients aged 65 years and older after regional anesthesia than after general anesthesia [19]. In the current study, 68.3% of patients in the open group underwent spinal anesthesia, whereas none of the patients in the SILTEP group were administered spinal anesthesia. Although the length of hospital stay was not significantly different between the two groups, we found that the urinary retention rates of the open group were significantly higher than those of the SILTEP group. Thus, SILTEP repair may have an advantage over open repair with spinal anesthesia by avoiding the development of postoperative urinary retention.

Previous studies suggest that laparoscopic inguinal hernia repair involves significantly lower postoperative pain and a shorter duration of recovery time compared with open repair [4,20]. In accordance with previous reports, our results demonstrated that the SILTEP group had significantly lower VAS scores and fewer analgesic requirements through postoperative day 1 than those in the open group. Recurrence rates in the SILTEP group were not significantly different from those in the open group. This finding indicates that SILTEP repair may have advantages for immediate postoperative pain. The shorter incision length in SILTEP repair than open repair may be an important reason for this difference. However, non-fixation of mesh might contribute to the reduction of pain at postoperative day 1 in the SILTEP group. Belyansky et al. demonstrated that the use of more than 10 tacks doubles the incidence of postoperative pain while having no effect on the rates of recurrence [21]. In a recent meta-analysis, without increasing the risk of early hernia recurrence, the non-fixation of mesh in laparoscopic inguinal hernia repair is comparable with tacker mesh fixation in terms of operation time, postoperative pain, postoperative complications, length of hospital stay, and chronic groin pain [22]. In this regard, the non-fixation of mesh in SILTEP repairs may result in lesser pain during the immediate postoperative period compared with that in open repair.

According to a recent meta-analysis, the operative time of unilateral SILTEP inguinal hernia repair varies from 40 to 90 minutes [5]. In the current study, the SILTEP group showed a mean operation time of 64.8 minutes, which was significantly longer than that of the open group. However, longer operative times were documented in early studies investigating single-incision laparoscopic cholecystectomy and are expected to occur during the learning curve inherent to the adoption of any new procedure [23]. Recently, single-incision laparoscopic surgery is being performed in various surgical fields such as cholecystectomy, appendectomy, colectomy, and gastrectomy [24]. Surgeons who already know the concept and ergonomics of single-incision laparoscopic surgery are increasing. Thus, we believe that the growing experience in other single-incision laparoscopic surgeries could shorten the operation time of SILTEP inguinal hernia repair. Such experience might also allow a one-step transition from open to SILTEP repair without prior CTEP repair experience.

The present study does have several limitations. First, we did not show data on cosmetic results, quality of life, and long-term outcomes because of a short follow-up period. Second, the study was designed retrospectively and various analyses for postoperative pain were not possible. Third, although propensity score matching reduced potential selection bias, the possibility of remaining unmeasured confounding variables may still be present. Thus, this may lead to biased results, and caution needs to be taken while interpreting these results. Finally, we did not analyze and compare the costs of SILTEP and open repair.

In summary, the current study demonstrated that the outcomes of SILTEP inguinal hernia repair were comparable to those of open repair. SILTEP repair may have advantages for reducing urinary retention and immediate postoperative pain compared to open repair. Future large-scale prospective controlled studies are needed to establish cosmetic results, quality of life, and long-term outcomes.

## Supporting information

**S1 Data. Anonymized data set.**
(XLSX)

## Author Contributions

**Conceptualization:** Kanghee Lee, Jin Ho Lee, Soomin Nam, Hyung Soon Lee.

**Data curation:** Kanghee Lee, Jin Ho Lee, Soomin Nam, Jae Uk Chong, Hyung Soon Lee.

**Formal analysis:** Jin Ho Lee, Soomin Nam, Jae Uk Chong.

**Writing – original draft:** Kanghee Lee, Jae Uk Chong, Hyung Soon Lee.

**Writing – review & editing:** Hyung Soon Lee.

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
