## [Decision Letter · Decision Letter 0]

8 Jan 2021

PONE-D-20-38169

Outcomes of Open Versus Single-Incision Laparoscopic Totally Extraperitoneal Inguinal Hernia Repair Using Propensity Score Matching

PLOS ONE

Dear Dr. Lee,

Thank you for submitting your manuscript to PLOS ONE. After careful consideration, we feel that it has merit but does not fully meet PLOS ONE’s publication criteria as it currently stands. Therefore, we invite you to submit a revised version of the manuscript that addresses the points raised during the review process.

Please revise accordingly. 

We look forward to receiving your revised manuscript.

Kind regards,

Academic Editor

PLOS ONE

Journal Requirements:

"This study was supported by a faculty research grant from the National

Health Insurance Service Ilsan Hospital (NHIMC 2019CR027)."

Reviewers' comments:

Reviewer's Responses to Questions

**Comments to the Author**

1. Is the manuscript technically sound, and do the data support the conclusions?

Reviewer #1: Yes

Reviewer #2: Yes

2. Has the statistical analysis been performed appropriately and rigorously? 

Reviewer #1: Yes

Reviewer #2: Yes

3. Have the authors made all data underlying the findings in their manuscript fully available?

Reviewer #1: Yes

Reviewer #2: No

4. Is the manuscript presented in an intelligible fashion and written in standard English?

Reviewer #1: Yes

Reviewer #2: Yes

5. Review Comments to the Author

Reviewer #1: It is a well written manuscript about an important new surgical technique of repair of inguinal hernia.

The language is well written in standard format

On first impression I would have believed that there is no need to compare single-incision technique with open repair rather than comparing with conventional laparoscopic hernia repairs, when already similar comparisons have been published with better results for laparoscopic procedures. but I would agree with the authors that there should be independent published data about comparing the single-incision technique with open repair.

The data analysis is well done.

Over all its a very well presented manuscript.

Reviewer #2: The article is a good read, clear, concise with good written English.

Ideally, evidence-based medicine should use prospective, randomized studies. These studies are difficult and expensive and are therefore rarely available and we resort to modified observational studies such as propensity score matching.

The most important limitation of propensity score methods is that although they can bring an equilibrium to the observed baseline covariates between the two groups mentioned in the study, they do nothing to evaluate and balance the unmeasured characteristics and confounders. Taking these aspects into consideration, unlike randomized control trials, propensity score analyses are limited in interpretation exactly due to these unmeasured confounding variables. This can lead to biased results and caution needs to be taken when the results are interpreted. The authors should mention these aspects in the study limitations.

Regarding the title, the authors should mention that this is a single institutional study.

The authors mention only 1 intraoperative incident in the SILTEP group, the bladder lesion. In three years of observation there were no conversions to TAP due to peritoneum rupture or conversion to open surgery?

Also the authors need to make available the data used for analysis of course by maintaining patient anonymity as to comply with the Plos One guidelines and to ensure the reproductivity of the results.

6. PLOS authors have the option to publish the peer review history of their article (what does this mean?). If published, this will include your full peer review and any attached files.

Reviewer #1: No

Reviewer #2: No

---

## [Author Response · Author response to Decision Letter 0]

12 Jan 2021

Review Comments to the Author

Reviewer #1: It is a well written manuscript about an important new surgical technique of repair of inguinal hernia. The language is well written in standard format On first impression I would have believed that there is no need to compare single-incision technique with open repair rather than comparing with conventional laparoscopic hernia repairs, when already similar comparisons have been published with better results for laparoscopic procedures. but I would agree with the authors that there should be independent published data about comparing the single-incision technique with open repair. The data analysis is well done. Over all its a very well presented manuscript.

Response: Thank you very much for the comments.

Reviewer #2: The article is a good read, clear, concise with good written English.

Ideally, evidence-based medicine should use prospective, randomized studies. These studies are difficult and expensive and are therefore rarely available and we resort to modified observational studies such as propensity score matching.

The most important limitation of propensity score methods is that although they can bring an equilibrium to the observed baseline covariates between the two groups mentioned in the study, they do nothing to evaluate and balance the unmeasured characteristics and confounders. Taking these aspects into consideration, unlike randomized control trials, propensity score analyses are limited in interpretation exactly due to these unmeasured confounding variables. This can lead to biased results and caution needs to be taken when the results are interpreted. The authors should mention these aspects in the study limitations.

Regarding the title, the authors should mention that this is a single institutional study.

The authors mention only 1 intraoperative incident in the SILTEP group, the bladder lesion. In three years of observation there were no conversions to TAP due to peritoneum rupture or conversion to open surgery?

Also the authors need to make available the data used for analysis of course by maintaining patient anonymity as to comply with the Plos One guidelines and to ensure the reproductivity of the results.

Response: 

- The following details on the limitations of propensity score matching was added on the discussion section:

 The present study does have several limitations. First, we did not show data on cosmetic results, quality of life, and long-term outcomes because of a short follow-up period. Second, the study was designed retrospectively and various analyses for postoperative pain were not possible. Third, although propensity score matching reduced potential selection bias, the possibility of remaining unmeasured confounding variables may still be present. Thus, this may lead to biased results, and caution needs to be taken while interpreting these results. Finally, we did not analyze and compare the costs of SILTEP and open repair.

- The title has been revised as requested as follows:

Outcomes of open versus single-incision laparoscopic totally extraperitoneal inguinal hernia repair using propensity score matching : a single institution experience

- There was only 1 patient with open conversion due to peritoneal tearing. Surgery with a single port is usually performed even if peritoneal tearing occurs during the operation. Peritoneal tears during surgery can significantly impair the progress of surgery, and the operation time is increased. However, when peritoneal tearing occurred, we performed peritoneal repair using intracorporeal laparoscopic suture, endoloop, and a hemolok clip. Thus, it was possible to continue the operation with a single port without open conversion or conversion to TAP.

- Data set was added to the revised manuscript on the supporting information section.

---

## [Decision Letter · Decision Letter 1]

15 Jan 2021

Outcomes of open versus single-incision laparoscopic totally extraperitoneal inguinal hernia repair using propensity score matching : a single institution experience

PONE-D-20-38169R1

Dear Dr. Lee,

We’re pleased to inform you that your manuscript has been judged scientifically suitable for publication and will be formally accepted for publication once it meets all outstanding technical requirements.

Kind regards,

Academic Editor

PLOS ONE

Additional Editor Comments (optional):

Reviewers' comments:

Reviewer's Responses to Questions

**Comments to the Author**

1. If the authors have adequately addressed your comments raised in a previous round of review and you feel that this manuscript is now acceptable for publication, you may indicate that here to bypass the “Comments to the Author” section, enter your conflict of interest statement in the “Confidential to Editor” section, and submit your "Accept" recommendation.

Reviewer #1: All comments have been addressed

Reviewer #2: All comments have been addressed

2. Is the manuscript technically sound, and do the data support the conclusions?

Reviewer #1: Yes

Reviewer #2: Yes

3. Has the statistical analysis been performed appropriately and rigorously? 

Reviewer #1: Yes

Reviewer #2: Yes

4. Have the authors made all data underlying the findings in their manuscript fully available?

Reviewer #1: (No Response)

Reviewer #2: Yes

5. Is the manuscript presented in an intelligible fashion and written in standard English?

Reviewer #1: (No Response)

Reviewer #2: Yes

6. Review Comments to the Author

Reviewer #1: Thank you very much for addressing the comments and issues raised by reviewers. I agree that the necessary amendments have been incorporated in the revised manuscript.

Reviewer #2: The authors have adressed the raised issues. The article is fit for publication from my own personal point of view.

7. PLOS authors have the option to publish the peer review history of their article (what does this mean?). If published, this will include your full peer review and any attached files.

Reviewer #1: No

Reviewer #2: No

---

## [Editor Report · Acceptance letter]

20 Jan 2021

PONE-D-20-38169R1 

Outcomes of open versus single-incision laparoscopic totally extraperitoneal inguinal hernia repair using propensity score matching : a single institution experience 

Dear Dr. Lee:

I'm pleased to inform you that your manuscript has been deemed suitable for publication in PLOS ONE. Congratulations! Your manuscript is now with our production department. 

Kind regards, 

on behalf of

Dr. Robert Jeenchen Chen 

Academic Editor

PLOS ONE